# Acousto-Optic Cells with Phased-Array Transducers and Their Application in Systems of Optical Information Processing

**DOI:** 10.3390/ma14020451

**Published:** 2021-01-18

**Authors:** Vladimir Balakshy, Maxim Kupreychik, Sergey Mantsevich, Vladimir Molchanov

**Affiliations:** 1Faculty of Physics, Lomonosov Moscow State University, 119991 Moscow, Russia; info@physics.msu.ru (M.K.); np@ntcup.ru (S.M.); 2Scientific and Technological Centre of Unique Instrumentation of the Russian Academy of Sciences, 117342 Moscow, Russia; 3Acousto-Optical Research Center at National University of Science and Technology MISIS, 119049 Moscow, Russia; aocenter@mail.ru

**Keywords:** acousto-optics, phased-array piezoelectric transducers, acousto-optic materials, anisotropic Bragg diffraction, acousto-optic devices

## Abstract

This paper presents the results of theoretical and experimental studies of anisotropic acousto-optic interaction in a spatially periodical acoustic field created by a phased-array transducer with antiphase excitation of adjacent sections. In this case, contrary to the nonsectioned transducer, light diffraction is absent when the optical beam falls on the phased-array cell at the Bragg angle. However, the diffraction takes place at some other angles (called “optimal” here), which are situated on the opposite sides to the Bragg angle. Our calculations show that the diffraction efficiency can reach 100% at these optimal angles in spite of a noticeable acousto-optic phase mismatch. This kind of acousto-optic interaction possesses a number of interesting regularities which can be useful for designing acousto-optic devices of a new type. Our experiments were performed with a paratellurite (TeO_2_) cell in which a shear acoustic mode was excited at a 9° angle to the crystal plane (001). The piezoelectric transducer had to nine antiphase sections. The efficiency of electric to acoustic power conversion was 99% at the maximum frequency response, and the ultrasound excitation band extended from 70 to 160 MHz. The experiments have confirmed basic results of the theoretical analysis.

## 1. Introduction

The basis of modern acousto-optic (AO) devices, applied in systems of optical information processing, is an AO cell that is usually made of a crystal with a certain cut. Ultrasonic waves are excited by piezoelectric plates attached to one of the cell faces. In these instruments, a type of light diffraction by ultrasound close to the Bragg regime is usually used [1,2,3]. This regime is realized at sufficiently high acoustic frequencies (usually over 100 MHz) and allows us to create devices with good performance and low light loss. However, as a disadvantage, this type of interaction has high selectivity, which means the AO effect magnitude has a strong dependence on the frequency of ultrasound *f*, the angle of light incidence $\theta_{0}$ and the wavelength of optical radiation λ. An important characteristic of the effect is the Bragg angle $\theta_{B}$. When the light beam falls at a Bragg angle (the phase-matching condition), the most effective light scattering into the diffraction order occurs, and a decrease in the diffraction efficiency by 3 dB determines the boundary of the AO interaction region.

In principle, the interaction region can be extended by reducing the length of AO interaction *L* (the width of the acoustic beam in the direction of light propagation), but this will result in a reduction of the diffraction efficiency and/or an increase in the acoustic power required for obtaining a necessary diffraction efficiency. Taking into consideration this negative situation, A. Korpel et al. [4] proposed the use of a step-sectional piezoelectric transducer for the excitation of an acoustic beam with a rotating radiation pattern. In their AO cell, the acoustic wavefront rotated with ultrasound frequency, adjusting to the optimal angle of light incidence.

However, such a cell was very difficult to manufacture, so planar structures were then proposed [5,6,7,8,9]. A planar structure is shown in Figure 1; in the first variant (Figure 1a), one piezoelectric plate (shown in green) is used, and separate sections are formed by partitioning the internal and external electrodes. In another case (Figure 1b), the internal electrode is first made solid, and then grooves are sawn at the final stage. The pictures show the AO cells with transducers containing four sections that are connected electrically in series, but in such a way that the direction of the electric field (shown by the arrows) in the adjacent sections is opposite. Due to this, acoustic beams from each section are excited with a phase shift of π, and a system of equivalent wavefronts (shown by two dashed lines) is formed, turned by an angle Ψ relative to the transducer plane. This angle is determined by the expression:(1)$$\Psi =\pm \frac{\Lambda }{2d}=\pm \frac{V}{2fd}$$
where Λ=V/f is the acoustic wavelength, *V* is its velocity and *d* is the period of the transducer array.

In comparison with a homogeneous (nonsectioned) transducer, here, the transfer function of the AO cell contains two main maxima, which are located symmetrically with respect to the Bragg angle $\theta_{B}$, as shown in Figure 2 with the red curve. It follows from the figure that AO diffraction is absent when the light beam falls at the Bragg angle ($\theta_{0}=\theta_{B}$). This is due to the fact that partial diffracted waves generated in neighboring acoustic beams are phase-shifted by π and therefore dampen each other during interference. However, there are other maxima situated equidistantly with the period Λ/d, which are inscribed into the dashed green curve showing the radiation pattern of a separate section of the transducer. The width of the maxima is equal to Λ/md, where *m* is the number of periods of the transducer array. Thus, we can conclude that in the case of the phased-array transducer, the concept of the Bragg angle as the angle of light incidence, at which the phase-matching condition is fulfilled and the maximum diffraction efficiency is observed, is incorrect. Here, we can talk about the optimal angles of light incidence $\theta_{opt}=\theta_{B}\pm \Lambda /2d=\theta_{B}\pm V/2fd$, which correspond to the maximal scattering of light, although the phase-matching condition is violated. It can be noticed that these optimal angles conform to the conventional Bragg angles when light falls on the equivalent wavefronts. Consequently, such a structure of the acoustic field can be considered as a superposition of two fields excited by two solid piezoelectric transducers rotated relative to each other by the angle Λ/d. Our calculations show that the diffraction efficiency here can reach 100%, despite a noticeable phase mismatch. However, this requires slightly higher acoustic power [10,11].

As follows from Equation (1), when the ultrasound frequency decreases, the main lobes of the transducer radiation pattern diverge. Therefore, if one chooses the true angle of light incidence, the equivalent wavefront will rotate, adjusting to the changing Bragg angle. This adjustment will not be complete since the Bragg angle depends on the frequency linearly, but the angle of rotation Ψ is in accordance with the hyperbolic law. The best correction of the Bragg angle is obtained when $d\theta_{opt}/df=0$. This condition is satisfied at the frequency:(2)$$f^{*}=V\sqrt{\frac{n}{\lambda d}}$$
where *n* is the refractive index of the AO medium and λ is the optical wavelength in vacuum. Thus, by selecting an appropriate period *d* of the phased-array transducer, we can set the operating point $f^{*}$ in any desired frequency range.

The interest in phased-array transducers weakened noticeably after anisotropic AO diffraction entered the practice of acousto-optics [12]. The main advantage of anisotropic diffraction is that it consists of a significantly more complicated frequency dependence on the Bragg angles in comparison with the AO interaction in an isotropic medium [1,2,3]. This feature makes it possible to choose optimal interaction geometry for each separate AO device. For example, for AO deflectors, the optimal area is that where $d\theta_{B}/df\to 0$, whereas for AO video-filters, it is the area where $d\theta_{B}/df\to \infty$.

The use of the phased-array transducers in combination with anisotropic diffraction gives more complicated types of Bragg curves, which open up new opportunities for improvement of AO device characteristics [13,14]. Therefore, the aim of this work is to study the peculiarities of anisotropic AO interactions in an acoustic field created by the phased-array transducers with antiphase excitation of adjacent sections.

## 2. Acousto-Optic Effect in the Field of Phased-Array Transducer

Figure 3 illustrates the problem in the most general case of an anisotropic medium in which acoustic beams propagate with a walk-off angle α. We assume that the inclined phase grating created by the acoustic wave occupies the area of space between the infinite planes x=0 and x=L. The wave vector of ultrasound is inclined at an angle α. The width of each acoustic column is *l*. Therefore, the second column occupies the space between the planes x=l+a and x=2l+a, where *a* is the gap between the acoustic beams, etc. Thus, the period of the transducer structure is $d=l+a=l\left(1+\xi \right)$. The initial acoustic phase in the first column is equal to Φ=0, and in the following column, is Φ=π; at that time the phase shift between the adjacent beams is equal to ΔΦ=π. The total number of beams is *m.*

The regime of anisotropic diffraction with Bragg scattering of light into two diffraction orders, zero and +1st or zero and –1st, is considered here for this structure. In this case, the calculation has to take into account two optical plane waves. The first wave is falling; it is characterized by the wave vector $k_{0}$ and the incidence angle $\theta_{0}$. The second one is a diffracted wave with the wave vector $k_{\pm 1}$ and the diffraction angle $\theta_{\pm 1}$. The interaction of these waves is described by the following system of equations [15]:(3)$$\left\{\begin{array}{c} \frac{dC_{0}}{dX}=\pm \frac{\Gamma }{2}C_{\pm 1}\exp\left[\pm j\left(R_{\pm }X-\Phi \right)\right]\\ \frac{dC_{\pm 1}}{dX}=\mp \frac{\Gamma }{2}C_{0}\exp\left[\mp j\left(R_{\pm }X-\Phi \right)\right] \end{array}\right.$$
where, for the convenience of numerical calculations, dimensionless values are introduced: the normalized amplitudes of the incident $C_{0}$ and diffracted $C_{\pm 1}$ waves, the coordinate X=x/l, the Raman–Nath parameter
(4)$$\Gamma =\frac{2\pi l\Delta n}{\lambda \cos\theta_{0}}$$
and dimensionless phase mismatch
(5)$$R_{\pm }=\frac{2\pi }{\lambda }l\left[n_{0}\cos\theta_{0}\mp \frac{\lambda f}{V}\sin\alpha -\sqrt{n_{\pm 1}^{2}-{\left(n_{0}\sin\theta_{0}\pm \frac{\lambda f}{V}\cos\alpha \right)}^{2}}\right]$$

In these formulas, $n_{0}$ and $n_{\pm 1}$ are the refractive indices for incident and diffracted light, and Δn is the amplitude of the refractive index change under the action of the acoustic wave. A wave vector diagram of the AO interaction is shown in Figure 3 on the right; it corresponds to the vector relationship [1]:(6)$$k_{0}+K+\frac{R_{\pm }}{l}=k_{\pm 1}$$

Here, we have taken into consideration that the phase mismatch vector $R_{\pm }$ is perpendicular to the boundaries of the acoustic beams.

When the light waves propagate in the periodic acoustic field, the optical energy is redistributed between them. Our task is to find the amplitudes of the waves at the output of the structure: $C_{0}\left(L\right)$ and $C_{\pm 1}\left(L\right)$. This can be fulfilled by recording a recurrence relation connecting the input and output fields for the *i*-th acoustic beam. The boundary conditions at the input have the following form:(7)$$\left\{\begin{array}{l} E_{0}\left(x\right)=E_{i}C_{0}^{\left(i\right)}\exp\left[j\left(\omega_{o}t-k_{0}\cos\theta_{0}x-k_{0}\sin\theta_{0}z\right)\right]\\ E_{\pm 1}\left(x\right)=E_{i}C_{\pm 1}^{\left(i\right)}\exp\left[j\left(\omega_{\pm 1}t-k_{\pm 1}\cos\theta_{\pm 1}x-k_{\pm 1}\sin\theta_{\pm 1}z\right)\right] \end{array}\right.$$
where $E_{i}$ is the amplitude of the incident optical wave, $\omega_{0}$ and $\omega_{\pm 1}$ are the frequencies of incident and diffracted light, $\omega_{\pm 1}=\omega_{0}\pm \Omega$, and Ω=2πf is the cyclic frequency of the ultrasound. Solving Equation (3) with the boundary conditions of Equation (7), we obtain the expressions for the amplitudes at the input of the (*i* + 1)-st acoustic column:(8)$$C_{0}^{\left(i+1\right)}=\left[C_{0}^{\left(i\right)}\left(A\mp j\frac{R_{\pm }}{2}B\right)\pm C_{\pm 1}^{\left(i\right)}\frac{\Gamma }{2}B\exp\left(\mp j\Phi \right)\right]\exp\left(\pm j\frac{R_{\pm }}{2}\right)$$
(9)$$C_{\pm 1}^{\left(i+1\right)}=\exp\left(\mp j\frac{R_{\pm }}{2}\right)\left[C_{\pm 1}^{\left(i\right)}\left(A\pm j\frac{R_{\pm }}{2}B\right)\mp C_{0}^{\left(i\right)}\frac{\Gamma }{2}B\exp\left(\pm j\Phi \right)\right]\exp\left[\pm jR_{\pm }\left(1+\xi \right)\right]$$

The phase shift, introduced by the area of empty space between the acoustic columns, is then taken into account [16]:(10)$$A\equiv \cos\frac{\sqrt{\Gamma^{2}+R_{\pm }^{2}}}{2},\;B\equiv \sin c\frac{\sqrt{\Gamma^{2}+R_{\pm }^{2}}}{2\pi }$$

However, the equal phase shift $\exp\left[-jk_{0}l\cos\theta_{0}\left(1+\xi \right)\right]$ is omitted. In Equations (3)–(10), the upper sign corresponds to light scattering into the +1st order, while the lower sign corresponds to the −1st order.

In acousto-optics, the angles of incidence and diffraction are usually calculated from the front of the acoustic wave. In accordance with this, we introduce angles $\phi_{0}=\theta_{0}-\alpha$ and $\phi_{\pm 1}=\theta_{\pm 1}-\alpha$. The condition of phase matching $R_{\pm }=0$ determines the Bragg angle $\phi_{B}$:(11)$$C_{0}\left(l+a\right)=C_{0}\left(l\right),\;C_{\pm 1}\left(l+a\right)=C_{\pm 1}\left(l\right)\exp\left[\pm jR_{\pm }\left(1+\xi \right)\right]$$

Equation (11) indicates that the acoustic walk-off does not affect the AO phase matching. However, the parameters Γ and $R_{\pm }$ depend on the walk-off angle α, as well as the optimal angles $\theta_{opt}$. This results in a change in the AO interaction range and, consequently, in the diffraction characteristics [17].

## 3. Computation Results

This section presents the results of our calculations for an acoustic field created by a phased-array transducer in a paratellurite (TeO_2_) crystal. We used the original computer program developed on the MATLAB platform. This program takes into account the optical activity of the material as, for the considered diffraction variants, the optical beams propagate close to the optical axis of the crystal, and the optical activity can have a noticeable effect on diffraction characteristics. The calculations are performed for the crystallographic plane $\left(1\bar{1}0\right)$ when a shear acoustic wave propagates at an angle of χ=4° to the plane (001). For this acoustic mode, the velocity is $V=0.632\times 10^{5}$ cm/s, and the walk-off angle is α=35°.

Figure 4a demonstrates the frequency dependences of the Bragg angles for optical wavelength λ=0.633 µm. The four curves correspond to different polarizations of incident light (ordinary “*o*” or extraordinary “*e*”) and to the scattering of light into the +1st or −1st orders of diffraction. For example, branch +*1e* characterizes the diffraction of the *e*-wave into the +1st order. Points D and T indicate areas optimal for AO deflectors and video-filters, respectively [1,2,3]. 

Figure 4b displays the AO interaction area separately for branch *+1e* in coordinates $\phi_{0}-f$. The calculations are carried out for the case of a homogeneous (nonsectioned) transducer with a width of *l* = 1 mm. The diffraction efficiency $\varsigma ={\left|C_{+1}\right|}^{2}$ is illustrated by the color scheme from zero (dark blue) to one (dark red). The values ζ=1 correspond to the Bragg angles. The fine structure in the picture is caused by the lateral maxima of the function $\sin c\left(•\right)$ in Equation (9). The horizontal sections of the pattern determine frequency characteristics of the AO interaction $\varsigma \left(f\right)$ for specified incidence angles of light, and the vertical sections determine angular characteristics $\varsigma \left(\phi_{0}\right)$. The shape of the area reproduced curve *+1e* is shown in Figure 4a, and its width is defined primarily by the size of the transducer *l*.

In the case of a phased-array transducer with antiphase excitation of adjacent sections, the situation changes cardinally. Figure 5 shows this change when the period of structure *d* decreases. In these calculations, the number of sections is chosen equal to *m* = 4, and the normalized gap between sections is ξ=1.

We can see from Figure 5 that, for large periods of the transducer structure, a common area of AO interaction splits into two symmetrical domains (Figure 5a). Then, an additional domain appears from the side of low ultrasound frequencies (Figure 5b). Thereafter, the left domains merge and form a common area with a very complex shape (Figure 5c). Finally, this area splits, and three nonoverlapping domains form (Figure 5d). It should be noted that such unusual dynamics of AO interaction areas differ significantly from the variant of a homogeneous (nonsectioned) acoustic field. This can be considered as an advantage of the phased-array transducers because the optimal geometry of interaction in the most convenient frequency range can be obtained when designing AO devices.

Of particular interest is the case of Figure 5c, which shows a completely unique AO interaction structure characterized by low angular and frequency selectivity. To evaluate this result, one has to take into account that the high selectivity of the Bragg diffraction in many AO devices is an interfering factor that leads to a decrease in the frequency and angular ranges of AO interaction, and, consequently, to a deterioration of characteristics of AO devices in resolution and operation speed. Figure 5c proves that by using the same AO cell, both collimated light beams and image-carrying beams can be processed in a wide frequency range.

Two special points on the Bragg curves, D and T, are pointed out in Figure 4a. They determine the optimal AO interaction geometry for deflectors and video-filters [1,2,3]. In the case of anisotropic diffraction in uniaxial crystals, these points never coincide. This means that different AO interaction geometries must be applied when creating deflectors and filters. Our research has shown that this problem can be solved by using phased-array transducers, as Figure 5c demonstrates.

Figure 6 displays the characteristics of this unusual AO geometry. In Figure 6a, the operating frequency $f_{0}$, which corresponds to the coinciding points D and T, is presented in the dependence of the crystal cut angle χ and the period of the transducer structure d. The divergence angle between the incident and diffracted beams δϕ (important characteristic for video-filters) increases with the acoustic frequency according to the formula $\delta \phi \approx \lambda f_{0}/nV$. Unfortunately, this positive dependence is accompanied by a decreasing AO figure of merit *M* (green curve in Figure 6a), which falls from a maximum value $M=1200\times 10^{-18}$ s^3^/g at χ=0° to $M=42\times 10^{-18}$ s^3^/g at χ=10° [1].

Figure 6b shows detailed characteristics related to the case $f_{0}=21$ MHz. The frequency dependences are constructed for different angles of light incidence: $\phi_{0}=-0.5{}^{\circ}$ (blue line), $\phi_{0}=-1{}^{\circ}$ (red line), and $\phi_{0}=-1.4{}^{\circ}$ (green line). It is seen that the frequency range $\Delta f_{1}=65$ MHz is confirmed to the angular range $\Delta \phi_{0}=0.4{}^{\circ}$ (between green and red curves), while the range $\Delta f_{2}=42$ MHz can be realized in the angular range $\Delta \phi_{0}=0.9{}^{\circ}$ (between the green and blue curves). At the same time, the frequency $f_{0}=21$ MHz corresponds to the angular range of the AO interaction $\Delta \phi_{0}=4.2{}^{\circ}$, which is 3.1 times greater than in the case of the unsectioned transducer.

The calculation of AO interaction areas for the variant of ordinary polarization of incident light (branch *+1o* in Figure 4a) gives a significantly different result. The phased-array transducer also leads to a splitting of the interaction area into two domains. However, the change of these domains when varying the period *d* of the transducer structure is greatly different in comparison with the variant discussed above: the region with extremely low angular and frequency selectivity does not appear at all, as seen in Figure 7.

The comparison of the AO interaction areas for the branches *+1e* and *+1o* allows us to note another interesting and practically important feature, which is fundamentally impossible in the case of AO diffraction in the homogeneous acoustic field. Figure 8 demonstrates the frequency dependences of the optimal angles of light incidence $\phi_{opt}$, combined for branches *+1e* and *+1o*. The graphs show that the curves, belonging to different diffraction orders *+1e* and *+1o*, intersect at the point M at the frequency *f* = 67 MHz (Figure 8a) or *f* = 134 MHz (Figure 8b). This means that, at these frequencies, the optical waves with ordinary and extraordinary polarizations have to diffract into the same diffraction maximum of the +1st order. Such AO interaction geometry makes it possible to modulate the intensity of unpolarized light beams without optical losses.

In this regard, it should be noted that the problem of AO modulation of unpolarized light has existed since the early 1960s and has not yet received an effective solution. The easiest solution is to search for cuts of crystals with close values of AO figure of merit *M* for both optical eigenmodes. A suitable material is lead molybdate (PbMoO_4_) [1]. However, this crystal is characterized by a relatively small AO figure of merit value $M=36\times 10^{-18}$ s^3^/g and, most importantly, has bad thermophysical properties. Modulators based on the paratellurite crystal with a longitudinal acoustic wave along the *Z-*axis are commercially available, but this cut of the crystal is characterized by a noticeable difference in the figure of merit for the ordinary and extraordinary waves: $M_{o}=30\times 10^{-18}$ s^3^/g and $M_{e}=22\times 10^{-18}$ s^3^/g. Besides, we may also mention an exotic scheme of an AO modulator with two multidirectional acoustic beams having different frequencies, which was studied in [18].

Contrary to that, we propose a significantly simpler variant of the modulator based on a paratellurite crystal with a phased-array transducer. In this device, the figure of merit coefficients for both optical eigenmodes are the same (due to the anisotropic type of diffraction) and equal to $M=800\times 10^{-18}$ s^3^/g (for χ=4°). The position of the operating point M in Figure 8 is determined by the period of the transducer structure *d*. Choosing the period of the phased array transducer, one can change the operating point in a wide frequency range. Similar variants of the modulator are possible for other cuts of the paratellurite crystal.

A similar idea can be implemented for creating AO deflectors intended for the scanning of unpolarized optical beams. Figure 9a illustrates this situation for the variant λ=1.5 µm, χ=2°, and m=8 and d=0.24 mm. Here, the dashed curves demonstrate frequency dependences of the Bragg angles for a nonsectioned transducer, whereas the AO interaction domains for the phased-array transducer are shown by red and blue colors. The yellow color represents the overlap area. Letter D shows the position of the deflector operating point at f=49 MHz and $\phi_{0}=3{}^{\circ}$.

Frequency characteristics of this deflector for the incidence angle $\phi_{0}=3{}^{\circ}$ are presented in Figure 9b. The two curves relate to different polarizations of the incident light. It can be seen that the common frequency band is Δf=35 MHz. This means that such a deflector can provide optical beam scanning in an angular range of $\Delta \phi =\left(\lambda /nV\right)\Delta f=2.1{}^{\circ}$ with spatial resolution $N=\left(w/V\right)\Delta f=570$ resolvable points when the optical aperture of the AO cell is w=1 cm.

## 4. Experimental Results

Experimental studies were carried out with an AO cell made of a paratellurite crystal in the variant of Figure 1b. An acoustic wave in the form of a slow shear mode was excited in the crystallographic plane $\left(1\bar{1}0\right)$ at an angle χ=9° to the direction [110]. For this crystal cut, the sound velocity was equal to $V=0.69\times 10^{5}$ cm/s, and the walk-off angle was α=52.5°. A phased-array transducer was made of an *X*-cut lithium niobate crystal with an electromechanical coupling coefficient k=68%. The transducer had *m* = 9 sections with a width of *l* = 2 mm each and a relative gap between the sections of ξ=0.2 (Figure 10a). Thus, the total length of the transducer in the direction of light propagation was *L* = 2.12 cm. To effectively excite ultrasound, an RF generator was matched with the transducer with the help of reactive elements—an inductance coil and a capacitor. The efficiency of electric to acoustic power conversion in the maximum of the frequency characteristic was 99%, and the ultrasound excitation band extended from 70 to 160 MHz.

Figure 10 shows the angular characteristics of AO interaction (normalized diffraction efficiency $\zeta /\zeta_{\max}$ as a function of the incidence angle $\phi_{0}$) obtained at the acoustic frequency *f* = 105 MHz. The measurements were fulfilled at the scattering of the ordinary polarized optical radiation with a wavelength of λ=0.53 µm into the +1st diffraction order. The red curve relates to the case where all nine sections are connected in series. In good agreement with calculations, the angular characteristic consists of two main maxima located symmetrically on different sides of the Bragg angle. Decreasing the number of sections results in a broadening of the maxima and a reduction in their intensity. In our experiment, for the convenience of comparing the results, the ultrasound power was adjusted inversely proportional to the number of sections.

## 5. Conclusions

In this paper, we present the results of a study of anisotropic Bragg diffraction of light in a spatially periodical acoustic field created by a sectioned piezoelectric transducer with antiphase excitation of adjacent sections. The problem of AO interaction in this periodical acoustic field is solved, taking into account the strong optical and acoustic anisotropy of the interaction medium that is typical for many crystals used in modern acousto-optics. Numerical calculations are performed for an AO cell made of paratellurite crystal with the shear acoustic mode propagating at an angle χ=4° to the plane (001) of the crystal. It is shown that the AO interaction in such a structure is absent when the optical beam falls at the Bragg angle. However, angles of incidence exist, called “optimal”, at which the diffraction efficiency can reach 100%, despite the violation of the phase-matching condition. The areas of interaction for different periods of the transducer structure are then calculated. A number of unusual regularities of AO scattering of light are established, which can be useful in the development of a new type of AO devices, such as modulators, deflectors and filters [1,2,3], as well as dual-channel monochromators for stereoscopic research [19,20]. In particular, the possibility of implementing AO modulators and deflectors for controlling nonpolarized light is shown, which could provide significantly better characteristics than the currently known devices. Preliminary experimental studies are performed with a paratellurite cell, with section numbers varying from nine to two. Results of the experiment are in a good agreement with the theoretical analysis.

## Figures and Tables

**Figure 1 materials-14-00451-f001:**
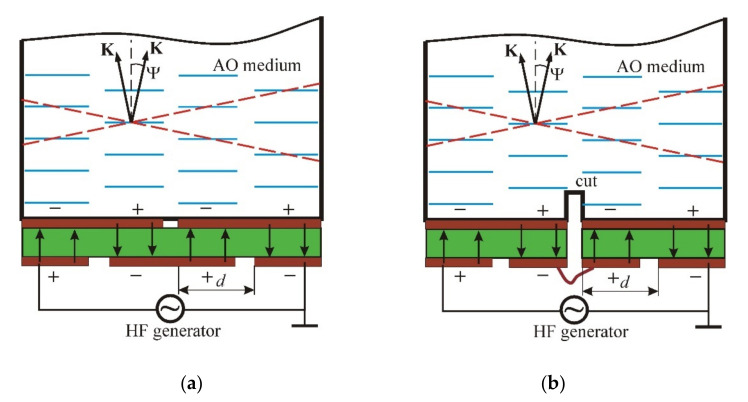
Two variants of flat phased-array transducers with antiphase excitation of adjacent sections: (**a**) Sectioning by partitioning the internal and external electrodes; (**b**) Sectioning by means of transducer cuts.

**Figure 2 materials-14-00451-f002:**
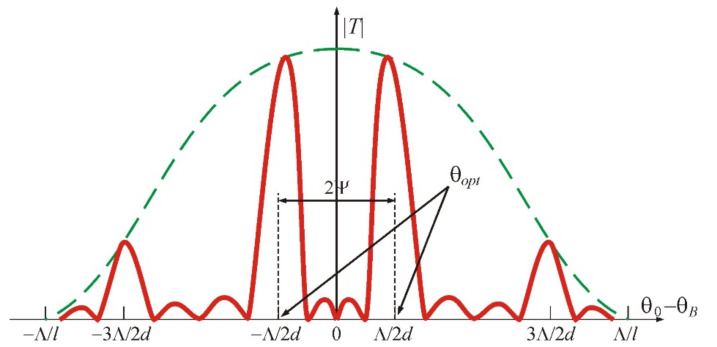
Radiation patterns of four-sectioned (red line) and one-sectioned (green line) transducers.

**Figure 3 materials-14-00451-f003:**
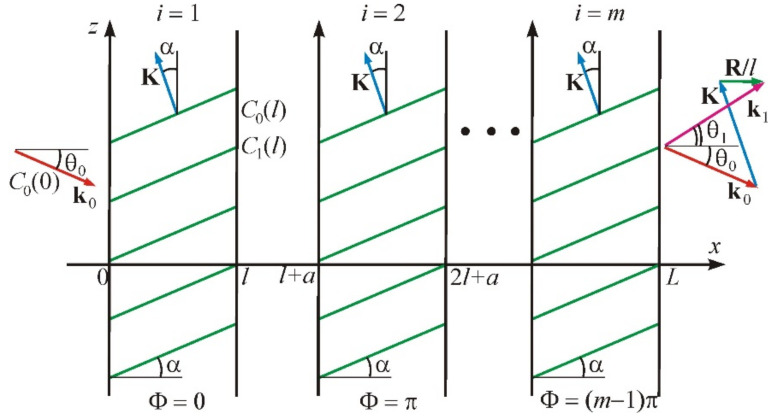
Statement of the problem of acousto-optic (AO) interaction in the field of the phased-array transducer.

**Figure 4 materials-14-00451-f004:**
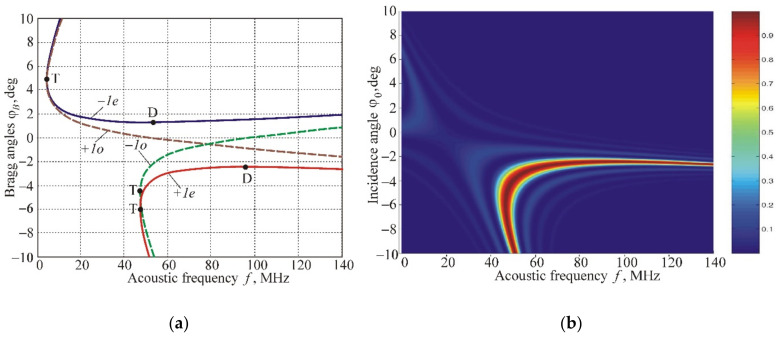
AO interaction in a paratellurite crystal: (**a**) Frequency dependencies of the Bragg angles in the case of light scattering in the +1st and –1st diffraction orders at different polarizations of incident optical radiation; (**b**) The area of AO interaction when the optical beam with extraordinary polarization diffracts into +1st order.

**Figure 5 materials-14-00451-f005:**
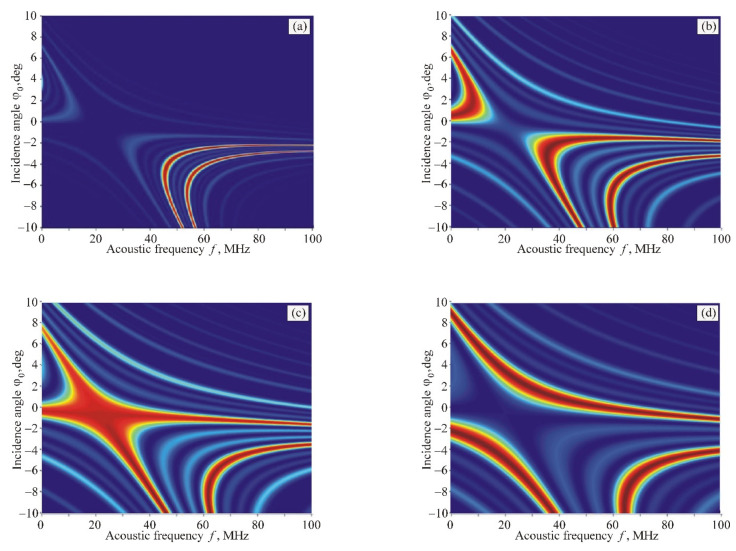
AO interaction areas for branch *+1e* in the case of *m* = 4 and different width of individual sections: (**a**) *l* = 0.4 mm; (**b**) *l* = 0.15 mm; (**c**) *l* = 0.11 mm; (**d**) *l* = 0.07 mm.

**Figure 6 materials-14-00451-f006:**
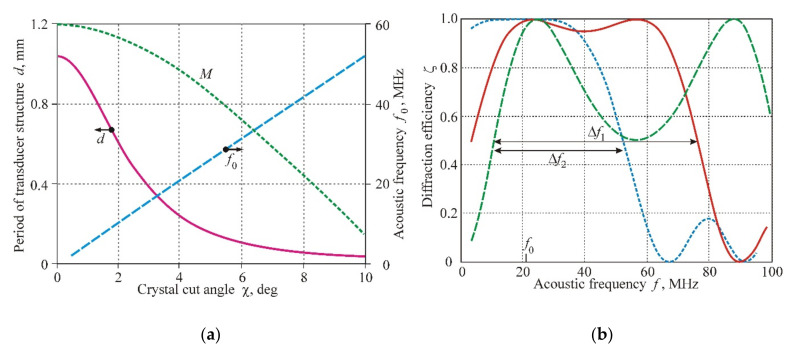
Characteristics of low-selective area shown in Figure 5c: (**a**) Operating frequency $f_{0}$, period of transducer structure *d* and figure of merit *M* as functions of crystal cut angle χ; (**b**) Frequency characteristics of AO interaction at the operating frequency $f_{0}=21$ MHz (χ=4°).

**Figure 7 materials-14-00451-f007:**
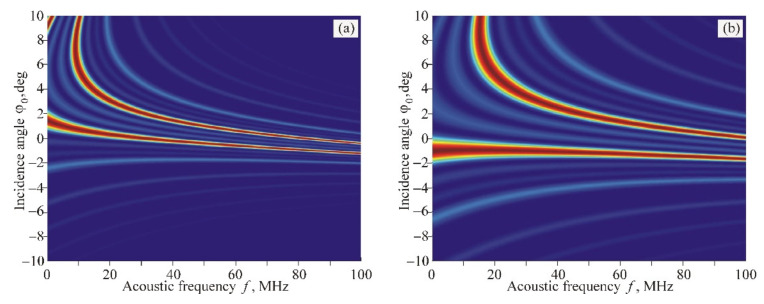
AO interaction areas for branch *+1o* in the case of *m* = 4 and different width of individual sections: (**a**) *l* = 0.15 mm; (**b**) *l* = 0.07 mm.

**Figure 8 materials-14-00451-f008:**
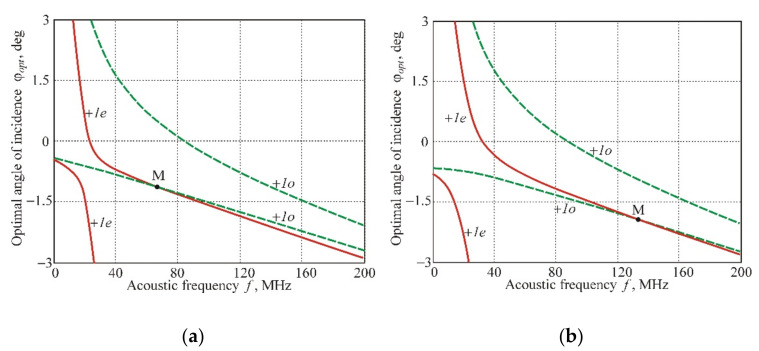
Frequency dependences of optimal angles for branches *+1е* and *+1о*: calculation for (**a**) *d* = 0.22 mm, and (**b**) *d* = 0.2 mm.

**Figure 9 materials-14-00451-f009:**
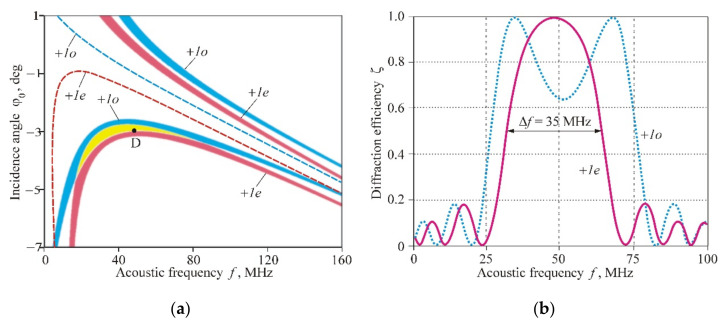
AO deflector of nonpolarized light: (**a**) Combined areas of AO interaction for +*1e* (red color) and +*1o* (blue color) diffraction branches, the overlap area is shown by yellow color; (**b**) Frequency characteristics of the AO deflector in the case of unpolarized light.

**Figure 10 materials-14-00451-f010:**
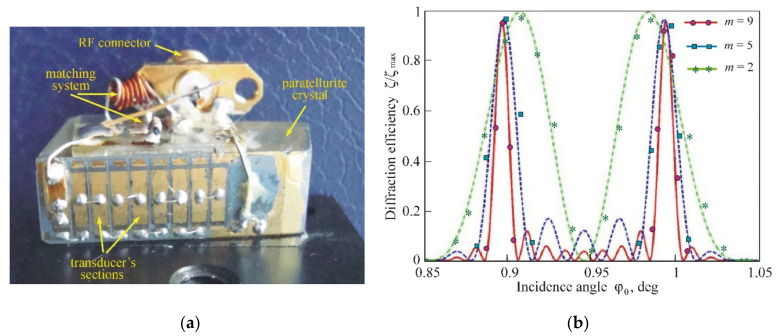
Experimental results: (**a**) AO cell with a nine-section transducer; (**b**) Angular characteristics of the AO diffraction for different number of connected sections: *m* = 9 (red), 5 (blue) and 2 (green).

## Data Availability

Data sharing is not applicable to this article.

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
