# Peer review of "Acousto-Optic Cells with Phased-Array Transducers and Their Application in Systems of Optical Information Processing"

_materials, 2021, doi:10.3390/ma14020451_

Round 1
Reviewer 1 Report
The authors present a study on a particular design of AO cells with phased-array transducers for applications in optical communication. The work is interesting and well written, so it can be accepted for publication in this journal.
There are some minor changes that are needed prior the publication of the manuscript, and these are listed in the followings.
In Figure 1 it is clearer to identify the two different curves only with the colours and not with the numbers. So I suggest to remove the number 1 and 2 from the graph, and refer to curves as the one in red and/or in green. E.g. at line 66, replace "... as shown in Figure 2 with curve 1" with "... as shown in Figure 2 with the red curve". In the Figure 2 legend please write "Figure 2. Radiation patterns of four-sectioned (red line) and one-sectioned (green line) transducers." instead of "Figure 2. Radiation patterns of four-sectioned (1) and one-sectioned (2) transducers."
The same suggestion comes for Figure 6b: please, use only the colours for distinguishing the three curves, not the numbers. Also add a short explanation for the differences of the three curves in the legend.
At line 309 it is written radiaction instead of radiation, please correct.
Author Response
Point 1: In Figure 1 it is clearer to identify the two different curves only with the colours and not with the numbers. So I suggest to remove the number 1 and 2 from the graph, and refer to curves as the one in red and/or in green. E.g. at line 66, replace "... as shown in Figure 2 with curve 1" with "... as shown in Figure 2 with the red curve". In the Figure 2 legend please write "Figure 2. Radiation patterns of four-sectioned (red line) and one-sectioned (green line) transducers." instead of "Figure 2. Radiation patterns of four-sectioned (1) and one-sectioned (2) transducers."
Response 1: All corrections pointed out here are made (see Figure 2 and lines 66, 84.
Point 2: The same suggestion comes for Figure 6b: please, use only the colours for distinguishing the three curves, not the numbers. Also add a short explanation for the differences of the three curves in the legend.
Response 2: All corrections are made as well (see Figure 6b and lines 239, 217-223.
Point 3: At line 309 it is written radiaction instead of radiation, please correct.
Response 3:The correction is made (see line 313).
Reviewer 2 Report
The authors explored anisotropic Bragg diffraction of light in a spatially periodical acoustic field created by a sectioned piezoelectric transducer with antiphase excitation of adjacent sections. The quality of the presentation is good.
I recommend the work can be published in MPDI provided the following few points were properly incorporated into the manuscript.
The authors should mention which software/numerical tools/values they used for numerical calculations in further detail.
Experimental results should be discussed in further detail and better connected to the previous data.
It would be much better to include some more recent references in the paper. In the current draft, there is only one reference after 2017.
I also felt the text font size was not the same throughout the entire manuscript; for example lines 265,266 and 267 look different. Those minor issues should also be resolved.
Author Response
Point 1: The authors should mention which software/numerical tools/values they used for numerical calculations in further detail.
Response 1: The sentence is added on lines 155,156: "At that we used the original computer program developed on the MATLAB platform."
Point 2: Experimental results should be discussed in further detail and better connected to the previous data.
Response 2: Reviewers # 2 and # 3 noted the same drawback of the article: the small volume of experimental results. This remark is absolutely fair and is explained by one reason: the quarantine due to the coronavirus. We have a cell made specifically for our experiments, as well as a ready-made experimental installation with all the necessary electronic equipment. But we have been deprived of the opportunity to conduct experimental research for six months, because our laboratory is closed, all students of our University are transferred to on-line education regime. The most unpleasant thing is that according to optimistic estimates, the quarantine may last else for next six months. But we would not like to delay the publication of our very original results for another six months. We believe that the theoretical results presented in our article may be of interest to specialists conducting research in the field of optoelectronics. Therefore, we have paid great attention to new devices that can be created based on our analysis.
Point 3: It would be much better to include some more recent references in the paper. In the current draft, there is only one reference after 2017.
Response 3: The situation is such that the first publication on this area of research appeared only in 2016. This is our presentation at the conference (point 13 in the list of references). We are not familiar with publications of other researchers in this particular field.
Point 4: I also felt the text font size was not the same throughout the entire manuscript; for example lines 265,266 and 267 look different. Those minor issues should also be resolved.
Response 4: The font is set the same throughout the manuscript. Lines 268,269, 270 (and some others) look different because of the formula in the text. We can't сhange them.
Reviewer 3 Report
In this work the authors present a theoretical and experimental study of an acousto-optic device.
My comments are:
1) The the working principle is explained widely, but the experimental section is very poor. A description of the setup used and a more clear comparison with the theoretical results are missing;
2) The application fields are not adequately mentioned as the title suggests. I think that a broad overview of the use of the acousto-optic effect in optical communication systems is important to increase the scientific significance of the work;
3) Bibliography is deeply incomplete with too many self-citations.
Author Response
Point 1: The working principle is explained widely, but the experimental section is very poor. A description of the setup used and a more clear comparison with the theoretical results are missing.
Response 1: See Response 2 to Reviewer # 2.
Point 2: The application fields are not adequately mentioned as the title suggests. I think that a broad overview of the use of the acousto-optic effect in optical communication systems is important to increase the scientific significance of the work.
Response 2: Possible new applications of our devices are mentioned on pages 205-209, 217-223, 243-251, 266-272, 278-288, 330-334. However, more detailed information is still required, including theoretical and experimental studies. Besides, we plan to patent these devices.
As for "broad overview of the use of the acousto-optic effect in optical communication systems", this material is well presented in monographs [1-3].
Point 3: Bibliography is deeply incomplete with too many self-citations.
Response 3: We cannot agree with this statement. The total number of publications on the acousto-optic effect contains several thousand. We have presented in the list of publications only those articles that are directly related to our research. The self-citation is now 40%. Is it too many? See please also our Response 3 to Reviewer # 2.
Round 2
Reviewer 3 Report
The authors did not accept any suggestions. Then, I leave the decision to the editor.